## How do people with dementia use the ambulance service? A retrospective study in England: the HOMEWARD project

Sarah Voss,[1] Janet Brandling,[1] Hazel Taylor,[2] Sarah Black,[3] Marina Buswell,[4] Richard Cheston,[1] Sarah Cullum,[5] Theresa Foster,[6] Kim Kirby,[3] Larissa Prothero,[6] Sarah Purdy,[7] Chris Solway,[8] Jonathan Richard Benger[1,2]

For numbered affiliations see end of article.

**Correspondence to**
Dr Sarah Voss;
sarah.voss@uwe.ac.uk

### ABSTRACT

**Objectives** An increasing number of older people are calling ambulances and presenting to accident and emergency departments. The presence of comorbidities and dementia can make managing these patients more challenging and hospital admission more likely, resulting in poorer outcomes for patients. However, we do not know how many of these patients are conveyed to hospital by ambulance. This study aims to determine: how often ambulances are called to older people; how often comorbidities including dementia are recorded; the reason for the call; provisional diagnosis; the amount of time ambulance clinicians spend on scene; the frequency with which these patients are transported to hospital.

**Methods** We conducted a retrospective cross-sectional study of ambulance patient care records (PCRs) from calls to patients aged 65 years and over. Data were collected from two ambulance services in England during 24 or 48 hours periods in January 2017 and July 2017. The records were examined by two researchers using a standard template and the data were extracted from 3037 PCRs using a coding structure.

**Results** Results were reported as percentages and means with 95% CIs. Dementia was recorded in 421 (13.9%) of PCRs. Patients with dementia were significantly less likely to be conveyed to hospital following an emergency call than those without dementia. The call cycle times were similar for patients regardless of whether or not they had dementia. Calls to people with dementia were more likely to be due to injury following a fall. In the overall sample, one or more comorbidities were reported on the PCR in over 80% of cases.

**Conclusion** Rates of hospital conveyance for older people may be related to comorbidities, frailty and complex needs, rather than dementia. Further research is needed to understand the way in which ambulance clinicians make conveyance decisions at scene.

## INTRODUCTION

Dementia is a progressive and irreversible condition resulting in a decline of cognitive, functional, behavioural and psychological abilities, and an associated loss of

### Strengths and limitations of this study

► This research examines the nature of ambulance service use by older people with dementia and comorbidities, and is the first study to describe this empirically.
► More than 3000 ambulance care records for patients aged over 65 years were examined. Data extraction was systematic using a predetermined protocol.
► Data were collected on a number of key variables including presenting complaint, presence of dementia and other comorbidities, call cycle time and the frequency of conveyance.
► The study was retrospective and carried out in two ambulance services in England; this may limit the generalisability of the findings.
► It was not possible to differentiate non-conveyance decisions that were made due to a patient's refusal to attend hospital.

independent living and social interaction. The number of people living with dementia is steadily increasing. In 2013, there were estimated to be 815 827 people with dementia in the UK; 773 502 were aged 65 years or over. This represents 1 in every 14 of the population aged 65 years and over.[1]

An increasing number of older people are accessing ambulance services and accident and emergency (A&E) departments.[2] The demands on urgent and emergency healthcare services are well documented and widespread.[3] Dementia is associated with higher levels of comorbidity than in an age-matched population.[4 5] More than 90% of people living with dementia have another health condition.[6] As a result, dementia is associated with an increased risk of hospitalisation.[7 8] Notably, when comorbidities are adjusted for, people with dementia have a higher incidence of A&E attendance.[4] This means that where a patient's comorbidities include dementia,

A&E attendance and hospital admission are all increased, compared with patients with the same comorbidities not including dementia.[8] The 'conversion rate' (proportion of A&E attendances that become a hospital admission) is very high in this patient group.[9]

However, we do not know whether these patients who attend A&E and are subsequently admitted are conveyed to A&E by ambulance. There are approximately 8 million emergency ambulance calls and 20 million A&E attendances in the UK every year.[2] There is some anecdotal evidence that older people with dementia are more likely to be conveyed to hospital following an emergency ambulance call because ambulance clinicians cannot provide a full clinical assessment, or access alternative services that may be more suitable.[10] Reducing unnecessary conveyance to A&E could therefore lead to a reduction in both A&E utilisation and acute hospital admissions for older people with dementia.[9] This would be beneficial to patients and the wider health system. Patients with dementia tend to do poorly in acute hospital settings; cognitive impairment results in a reduced threshold for sensory overload and distress which can lead to disruptive behaviours and worse patient experience and outcomes, including high rates of discharge to care homes, readmission and death.[11–13] In addition, over 40% of unplanned admissions of those aged over 70 are for people living with dementia,[14] and it is estimated that 25% of hospital beds in the UK are occupied by people with dementia with significant consequences for patient flow and healthcare resources.[15]

The use of emergency ambulance services by older people with dementia is not well understood, and has been the subject of very little research. A review by Buswell et al[10] highlights the issue of 'inappropriate' calls, where an ambulance is called as the last resort or a 'safety net'. The authors identified recurrent themes including the absence of alternatives and a lack of integration in healthcare. However, these assumptions have yet to be tested empirically.

Before exploring the opportunity to research interventions to reduce ambulance conveyance for this patient group, there is a need to determine the extent to which people with dementia and other comorbidities use ambulance services and are subsequently conveyed to hospital. Buswell et al studied the records of ambulance calls to people aged over 75 years and found that dementia was recorded in 14.5% of care records, with an additional 7.0% containing details suggestive of dementia or cognitive impairment.[16] Approximately, 15% of this patient cohort resided in care homes.

The aim of this research was to determine how often ambulances are called to older people with comorbidities including dementia, and if these patients are more likely to be conveyed to hospital. We also sought to determine: the reason for the call; the provisional diagnosis; the amount of time ambulance clinicians spend on scene. Given the evidence available to date, we hypothesised that when a person with comorbidities including dementia

required care, the clinicians would spend more time on scene, and patients would be more likely to be conveyed to hospital when compared with patients without dementia.

## METHODS

This was a retrospective cross-sectional study of ambulance patient care records (PCRs) from calls to patients aged 65 years and over in the geographical areas covered by the West Division of South Western Ambulance Service NHS Foundation Trust (SWASFT) and East of England Ambulance Service NHS Trust (EEAST). Data were collected from each Trust during 24-hour periods in January 2017 and July 2017 (for EEAST) and 48-hour periods in the same months for SWASFT.

PCRs from calls to patients aged 65 years and over within the geographical area covered by SWASFT West Division were accessed via a bespoke query run on the Trust's electronic care system, and the anonymised data were sent to the research team. In EEAST, electronic versions of PCRs were securely stored and interrogated using computer software. Access to these records was overseen by the clinical audit teams for the respective Trusts.

Health Research Authority (HRA) approval (IRAS: 202449) was obtained prior to data collection. Approval from an ethics committee was not required because the project involved the collection and analysis of retrospective and anonymised patient data only.

The records were examined by two researchers using a standard template and the data recorded using a coding structure (online supplementary appendix A). The template and coding structure were determined by the study team in advance of data extraction by obtaining a pilot sample of 40 PCRs from a different time frame which were scrutinised for relevant data. This provided a guide for the researcher to obtain information on: the time of the call; the length of the call; the reason for the call; whether dementia and any other comorbidities were noted in any section or free-text area of the PCR (supplementary material available on request); whether the patient lived alone, was in their own home or a care home; whether the patient was conveyed to hospital; any further referrals that were made. Electronic data were stored in a secure area of the University of the West of England and the University of Bristol servers. In order to assess inter-rater reliability, a sample of 64 PCRs from EEAST was extracted and coded by two additional members of the research team with agreement assessed using the Kappa statistic.

As the study uses retrospective data from ambulance PCRs, for some items the completion rates were low. However, the key items such as whether conveyance occurred and the length of call out had good completion rates. For most items the assumption was made that the data were 'missing at random', and so those with missing data for the item in question were removed from the analysis of that item only. For this reason the number in the analysis (n) is different for each item reported. For

the comorbidities, it was assumed that if the comorbidity had not been reported, then the patient did not have it, since there was no option to record 'no' for the individual comorbidities in the PCR.

Percentages and means with 95% CIs were determined using recognised methods for calculating an estimate of the population SD and used to report the key results. 95% CIs for percentages/proportions were calculated using the exact binomial method. 95% CIs for means were calculated using standard methods assuming a normal distribution, which is appropriate given the large sample size. The analysis is exploratory, seeking to describe the nature of ambulance calls to patients with and without dementia, and therefore uses descriptive statistics only. Statistical hypothesis testing was not conducted due to the number of factors being considered and the risk of finding spurious statistically significant results. Moreover, as the clinical decisions made during call-outs are likely to be influenced by, for example, the crew, day of the week and time of the call, all of these unknown factors would need to be taken into account to make hypothesis testing valid.

## Patient and public involvement

This project was partnered by the Alzheimer's Society Research Network who were consulted on the draft protocol and gave feedback on the design, methodology, analysis and dissemination plans. They also provided specific and detailed advice on the proposed data fields. An experienced patient and public representative (CS) was a member of the study management group and coauthor for this paper and contributed to all aspects of the project from inception to dissemination. Two additional Patient and Public Involvement (PPI) representatives were members of the steering committee.

## RESULTS

In total, 3037 PCRs were extracted: 2311 from SWASFT and 726 from EEAST.

The kappa statistic demonstrated good reliability, 26 of 37 items had a kappa of 0.61 or more indicating substantial agreement, including the coding of cognitive state (kappa=0.683) and definite dementia diagnosis (kappa=0.947).

## Patient and call demographics

Dementia was recorded in the PCR in 421 cases (13.9%; 95% CI 12.7% to 15.1%). The comparator group consisted of 2567 (84.5%) cases where dementia was not recorded. Forty-nine (1.6%) of PCRs were excluded because it was not clear whether dementia was recorded or not. See table 1 for a summary of the patient demographics. The mean age of patients in the dementia group was 85.1 years (95% CI 84.4 to 85.7) and 80.7 years (95% CI 80.4 to 81.0) in the comparator group. Ambulance calls to people with dementia were more likely to be to a residential or nursing home (41.9%; 95% CI 35.6% to 48.3%) than in the comparator group (5.4%; 95% CI 4.3% to 6.7%), whereas calls to individuals living in their own home were more likely in the comparator group (81.0%; 95% CI 78.8% to 83.0%) than the dementia group (52.8%; 95% CI 46.4% to 59.2%).

## Presenting condition and provisional diagnosis

The reason for the call (presenting condition) was less likely to be related to a cardiac or respiratory problem in the dementia group (4.8%; 95% CI 3.0% to 7.3% and 5.8%; 95% CI 3.7% to 8.4%) than the comparator group (10.8%; 95% CI 9.6% to 12.1% and 11.2%; 95% CI 10.0% to 12.5%), but more likely to be for a fall in the dementia group (15.6%; 95% CI 12.2% to 19.4%) than in the comparator group (9.1%; 95% CI 8.0% to 10.3%)

| Table 1 | Summary of patient demographics | | | | | |
|---|---|---|---|---|---|---|
| | **Comparator** | | **Dementia** | | **Total*** | |
| | **n=2567** | | **n=421** | | **n=3037** | |
| | Mean | St. D | Mean | St. D | Mean | St. D |
| Age (years) | 80.7 | 8.7 | 85.1 | 7.0 | 81.4 | 8.6 |
| No of ambulance staff on scene (SWASFT) | 2.7 | 1.4 | 2.7 | 1.7 | 2.7 | 1.4 |
| Gender | n | % | n | % | n | % |
| Male | 1162 | 45.4 | 166 | 39.5 | 1347 | 44.5 |
| Female | 1396 | 54.6 | 254 | 60.5 | 1680 | 55.5 |
| Location† | n=1390 | % | n=246 | % | n=1665 | % |
| Home | 1126 | 81.0 | 130 | 52.8 | 1275 | 76.6 |
| Care home | 75 | 5.4 | 103 | 41.9 | 186 | 11.2 |
| Public place | 118 | 8.5 | 7 | 2.9 | 126 | 7.6 |
| Other | 71 | 5.1 | 6 | 2.4 | 78 | 4.7 |

*Includes the 49 patients excluded due to uncertainty as to whether dementia was recorded.
†1372 missing.
SWASFT, South Western Ambulance Service NHS Foundation Trust.

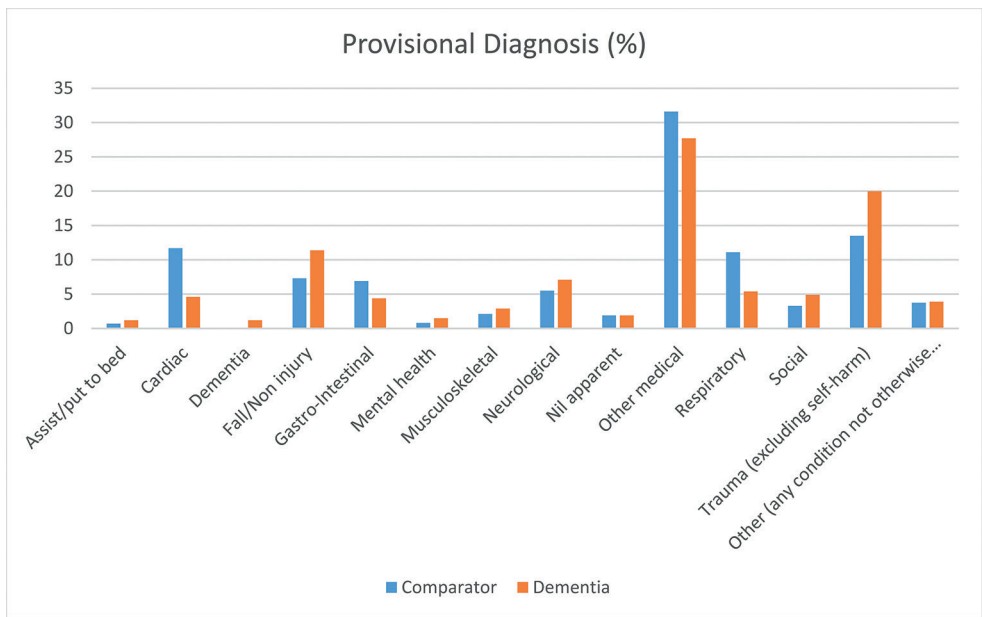

**Figure 1** Provisional diagnosis (n=2848: 411 with dementia and 2437 comparator group).

(figure 1). Similar findings were evident for the provisional diagnosis as recorded by the attending clinician. Patients with dementia were less often diagnosed with cardiac or respiratory problems (4.6%; 95% CI 2.8% to 7.1% and 5.4%; CI 3.4% to 8.0%) than patients in the comparator group (11.7%; 95% CI 10.4% to 13.0% and 11.1%; 95% CI 9.9% to 12.4%). Conversely, traumatic injuries and falls were more prevalent in the dementia group (20.0%; 95% CI 16.2% to 24.1% and 11.4%; 95% CI 8.5% to 14.9%) than in the comparator group (13.5%; 95% CI 12.2% to 14.9% and 7.3%; 95% CI 6.3% to 8.4%).

### Social circumstances and frailty

Patients with dementia were more likely to be living in a care home and to have a care package in place (figure 2).

Only 6.7% (95% CI 4.2% to 10.0%) of patients with dementia were recorded as living in their own home without a care package compared with 31.7% (95% CI 29.5% to 33.9%) of the comparator group. Similarly, 12.4% (95% CI 9.0% to 16.6%) of patients with dementia were living with extended family without a care package compared with 32.1% (95% CI 30.0% to 34.3%) of the comparator group. 25.8% (95% CI 21.0% to 31.0%) and 22.6% (95% CI 18.1% to 27.6%) of patients with dementia were living in a nursing or residential home compared with 3.5% (95% CI 2.7% to 4.5%) and 3.6% (95% CI 2.8% to 4.6%) of the comparator group.

Frailty scores were recorded by SWASFT in 1128 patients of which 70.2% (95% CI 63.1% to 76.6%) of patients

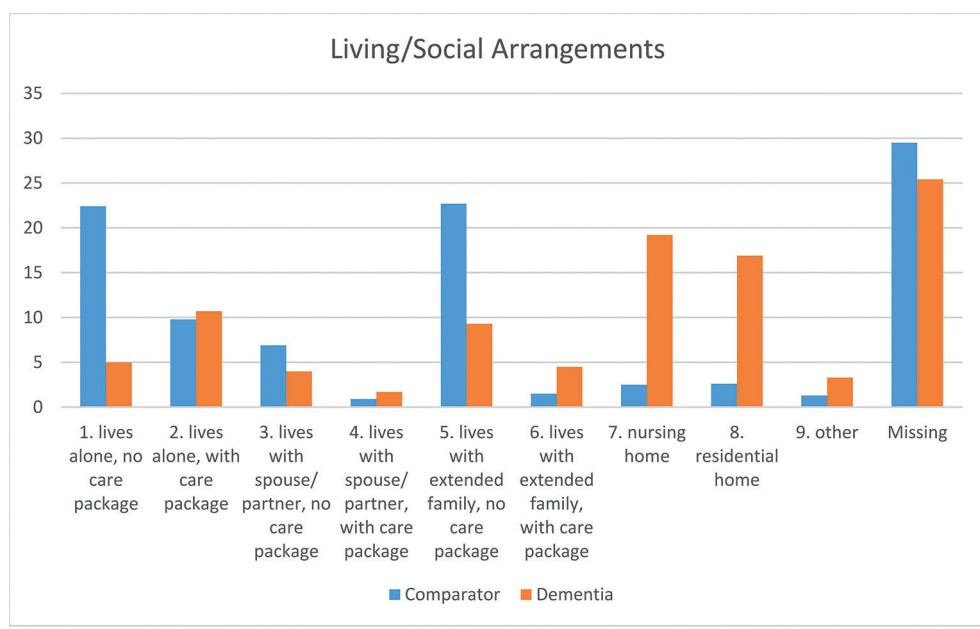

**Figure 2** Living and social arrangements.

| Table 2 | Frailty (n=1128 with frailty data) | | | |
| --- | --- | --- | --- | --- |
| | **Comparator group** | | **Dementia group** | |
| **Frailty score** | **N** | **%** | **N** | **%** |
| Very fit | 40 | 4.3 | 1 | 0.5 |
| Well | 137 | 14.6 | 2 | 1.1 |
| Managing well | 234 | 24.9 | 11 | 5.9 |
| Vulnerable | 111 | 11.8 | 18 | 9.6 |
| Mildly frail | 131 | 13.9 | 24 | 12.8 |
| Moderately frail | 147 | 15.6 | 43 | 22.9 |
| Severely frail | 108 | 11.5 | 80 | 42.6 |
| Very severely frail | 15 | 1.6 | 6 | 3.2 |
| Terminally ill | 17 | 1.8 | 3 | 1.6 |
| Total | 940 | 100 | 188 | 100 |

in the dementia group were assessed as moderately to severely frail compared with 30.5% (95% CI 27.6% to 33.6%) in the comparator group. Only 7.4% (95% CI 4.1% to 12.2%) of patients in the dementia group were assessed as managing well or being very fit compared with 43.7% (95% CI 40.5% to 47.0%) in the comparator group (table 2).

### Call cycle time and conveyance to hospital
The mean duration of the call (time of arrival on scene to time of closing the call) was shorter for people with dementia (85.2 min 95% CI 81.5 to 88.8) than for the comparator group (89.8 min 95% CI 88.2 to 91.5) (table 3). In 61.0% of all cases the patient was conveyed to hospital by ambulance. Patients in the dementia group were less likely to be taken to hospital (50.4%); 95% CI 45.5% to 55.2% compared with those in the comparator group (62.7%; 95% CI 60.8% to 64.6%). Fewer patients in the dementia group were recorded as experiencing pain (25.6%; 95% CI 21.5% to 30.1%) than in the comparator group (39.5%; 95% CI 37.6% to 41.4%).

### Comorbidities
One or more comorbidities were recorded for 2469 of the 3037 (81.3%; 95% CI 79.9% to 82.7%) cases. For patients in the dementia group, 352 of the 421 cases (83.6%; 95% CI 79.7% to 87.0%) had at least one comorbidity recorded, and for the comparator group it was 2076 of 2567 cases (80.9%; 95% CI 79.3% to 82.4%).

Respiratory and cardiac comorbidities were less likely to be recorded for patients with dementia, whereas neurological disorders, stroke and musculoskeletal conditions were more prevalent (figure 3). The presence of one or more comorbidities increased call cycle time and conveyance rate, irrespective of dementia (table 4).

### DISCUSSION
Dementia was recorded in the PCR in approximately 14% of cases. This finding is consistent with work carried out by Buswell et al,[16] which reported a rate of 14.5% in patients aged over 75 years.

Contrary to our hypotheses, patients aged over 65 years with dementia were less likely to be conveyed to hospital following an emergency call than patients over 65 years without dementia. In addition, the call cycle times were similar for patients regardless of whether or not they had dementia. This might indicate that the high A&E attendance rate for people with dementia observed in previous research[6 7] results from 'front door' attendances by people brought in by carers or relatives, rather than those conveyed to hospital by ambulance. Moreover, it is possible that because staff are aware of the negative outcomes associated with hospital admission for people with dementia, they avoid conveying these patients unless absolutely necessary.

One or more comorbidities were reported in the PCR in just over 80% of all cases, with little difference between the dementia group and those without dementia. This is incongruent with findings from other settings, where people with dementia have consistently been found to have a significantly higher incidence of comorbidity at hospitalisation[4 8] and in primary care.[5] This may be related to the fact that information recorded in ambulance records is obtained from sources on scene and is not linked reliably to other records of healthcare data. Reliable and accurate information about medical history

| Table 3 | Call cycle time and conveyance to hospital | | | | | |
| --- | --- | --- | --- | --- | --- | --- |
| | **Comparator** | | **Dementia** | | **Total** | |
| | **n=2567** | | **n=421** | | **n=3037** | |
| | Mean | St D | Mean | St D | Mean | St D |
| Call cycle time in min | 89.8 | 41.3 | 85.2 | 38.3 | 89.4 | 40.9 |
| | n | % | n | % | n | % |
| No conveyed* | 1598/2547 | 62.7 | 211/419 | 50.4 | 1838/3015 | 61.0 |
| No with pain | 1014 | 39.5 | 108 | 25.6 | 1132 | 37.3 |
| No with medical history† | 1697/1773 | 95.7 | 300/303 | 99.0 | 2024/2104 | 96.2 |

*22 missing.
†SWAST only (207 missing).
SWAST, South Western Ambulance Service NHS Foundation Trust.

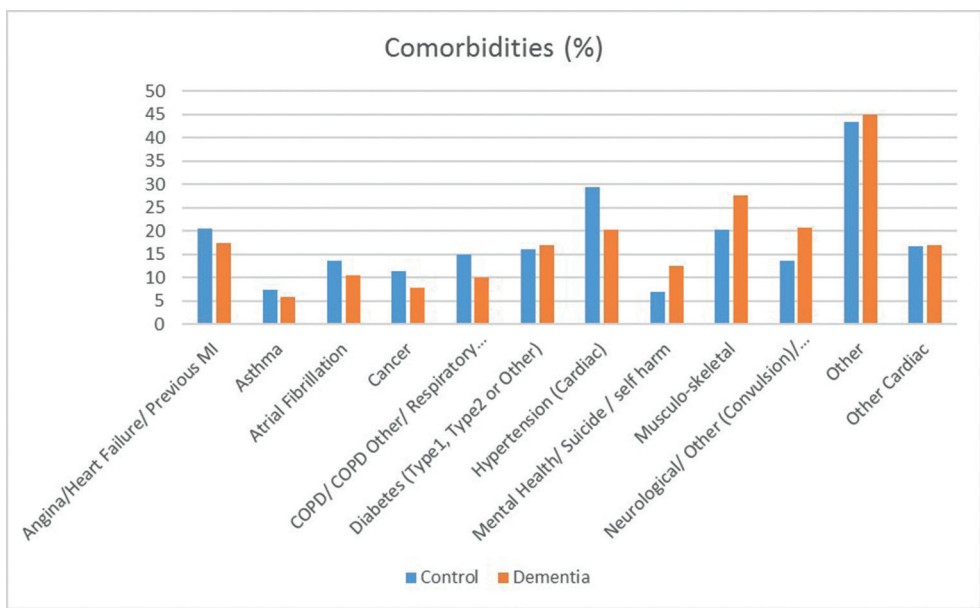

**Figure 3** Comorbidities recorded. COPD, chronic obstructive pulmonary disease; MI, myocardial infarction.

is not always available to ambulance staff. Nonetheless, when one or more comorbidities were recorded as present, patients were more likely to be conveyed to hospital. This difference was more evident in the comparator group than in the dementia group. In addition, call cycle times were longer for patients with comorbidities, regardless of the presence of dementia. It may therefore be deduced that call cycles last longer and conveyance rates are higher when older adults have complex needs or frailty associated with comorbidity; the presence of dementia is not necessarily an influencing factor in these outcomes.

Post hoc analyses of conveyance to hospital and call cycle time were calculated on patients in different age bands (65–74, 75–84 and 85+ years). The findings indicate that in general, for younger people the call cycle time is longer and they are more likely to be conveyed to hospital. This is consistent with previous research on the epidemiology of non-conveyed patients, which found that older patients account for the majority of these calls. Moreover, falls account for 34%–40% of all non-conveyed patients.[17 18]

Dementia is associated with an increased risk of falls, and an increased level of fall-related traumatic injury (31% vs 21%) was noted for this patient group. In previous research, Buswell et al[16] found falls to be the most common reason for an emergency call; in the present study, while a fall was the presenting complaint in over a quarter of cases overall, this proportion increased to almost half where dementia was recorded. Therefore, it is possible that the high number of falls in the dementia group accounts for the lower rate of conveyance. It was not possible in this research to establish which non-conveyance decisions were made by the attending clinician, and which were due to the patient's refusal to be transported, and this may also be a relevant factor. However, non-conveyance in older people who have fallen may be problematic since those not conveyed have been found to have a high rate of subsequent emergency healthcare contacts and an increased risk of death and hospital admission.[18] Although there is research ongoing to address prehospital assessment and management of people who have fallen,[19] it is important to consider the impact of dementia in these protocols.

| Table 4 | Hospital conveyance and call cycle time with comorbidities | | | | | |
|---|---|---|---|---|---|---|
| | **Conveyed to hospital** | | | **Call cycle time (min)** | | |
| | n | % | 95% CI | n | Mean | 95% CI |
| Total | 1838/3015 | 61.0 | (59.2% to 62.7%) | 2996 | 89.4 | (87.9 to 90.9) |
| Comparator | 1598/2547 | 62.7 | (60.8% to 64.6%) | 2532 | 89.8 | (88.2 to 91.5) |
| Comparator (no comorbidities) | 273/481 | 56.8 | (52.2% to 61.2%) | 480 | 82.8 | (79.1 to 86.6) |
| Comparator (with comorbidities) | 1325/2066 | 64.1 | (62.0% to 66.2%) | 2052 | 91.5 | (89.7 to 93.3) |
| Dementia | 211/419 | 50.4 | (45.5% to 55.2%) | 418 | 85.2 | (81.5 to 88.8) |
| Dementia (no comorbidities) | 33/69 | 47.8 | (35.6% to 60.2%) | 69 | 78.7 | (70.5 to 86.9) |
| Dementia (with comorbidities) | 178/350 | 50.9 | (45.5% to 56.2%) | 349 | 86.4 | (82.3 to 90.5) |

Of particular relevance to older people who have dementia, who have fallen and who are not conveyed is the assessment of pain; it is well recognised that pain is less often appreciated in dementia patients and is difficult to assess.[20 21] Despite the fact that we observed falls to be more common in the dementia group, pain was less often recorded in PCRs for those with dementia (25.6%) than for those in the comparator group (39.5%). More research is needed to establish reliable ways of assessing pain in people with dementia, to ensure that non-conveyance decisions are appropriate and do not lead to a further need for emergency healthcare.

Unsurprisingly, patients with dementia were more likely to be living in a care home (48%) than those without dementia (7%). Conveyance was more frequent for both groups when living in a care home than for those living in their own home; post hoc analysis indicated that when in a care home, people with dementia are still less likely to be conveyed to hospital (54.6%) than those without (66.9%). Therefore, it is possible that care home staff do not always feel able to accurately assess these patients, and may rely on ambulance staff who, in turn, need to be equipped and skilled to reliably identify pain and assess other symptoms in people with dementia.

There are several limitations that impact on the generalisability of the findings and the conclusions that can be drawn from our study. First, this was a retrospective study and there is the potential for bias that may affect the selection of comparator cases. However, great care was taken to ensure data extraction was robust and this is detailed in the coding template. Second, the samples were only extracted from two services over two short periods. The time periods were in January and July to capture any seasonal variation. Additional issues that may have affected the reliability and validity of the data were associated with dementia diagnosis; recording of dementia on the PCR does not equate to a definitive diagnosis. Conversely, it is likely that some of the comparator cases had undiagnosed or unrecognised dementia. Nonetheless, in the absence of robust evidence in this field, this study provides both useful information and indications for further research.

## CONCLUSION

The reason for a lower conveyance rate in patients recorded as having dementia remains unexplained. In this study, comorbidity was a more accurate predictor of conveyance to hospital than dementia per se. We have, however, shown that ambulance services are called to people with dementia for different reasons than people without dementia. Therefore, it is possible that the lower conveyance rate for dementia patients may be associated with non-injury falls or transient symptoms that are not perceived as requiring admission to hospital. However, unless pain can be reliably assessed, decisions on non-conveyance for people with dementia may not necessarily be appropriate. In addition, care and nursing staff may

have low thresholds for calling an ambulance for an individual who requires primary care management rather than conveyance to hospital. This would offer an explanation for the similar call cycle times we observed for both patients groups. However, further research is required to examine these issues in more detail.

This study has shown that higher rates of hospital admission for older people may be more related to complex needs and frailty than dementia. Given the unexpected nature of our findings, further research is required to understand how people with dementia interact with ambulance services, and the way in which ambulance clinicians make decisions at scene. There is much evidence to suggest that hospital admission for people with dementia results in adverse patient outcomes, and should be avoided where possible. However, evidence also suggests that patients who are not conveyed following a fall are at risk of poorer outcomes. Research on the assessment of pain and injury following a fall in people with dementia is a high priority, to ensure the most appropriate conveyance decisions are made.

**Author affiliations**
[1]Faculty of Health and Applied Sciences, University of the West of England, Bristol, UK
[2]Research Design Service, University Hospitals Bristol NHS Foundation Trust, Bristol, UK
[3]Research and Audit Department, South Western Ambulance Service NHS Foundation Trust, Plymouth, UK
[4]Centre for Research in Public Health and Community Care, University of Hertfordshire, Hatfield, UK
[5]Faculty of Medical and Health Sciences, University of Auckland, Auckland, New Zealand
[6]Research Support Services, East of England Ambulance Service NHS Trust, Melbourn, UK
[7]School of Social and Community Medicine, University of Bristol, Melbourn, UK
[8]Research Network, Alzheimer's Society, London, UK

**Acknowledgements** The authors would like to thank members of the study steering committee for their contribution to this research: Steve Iliffe (Professor of Primary Care for Older People, University College London); Philp Bath (The Stroke Association Professor of Stroke Medicine, Chair and Head of the Division of Clinical Neuroscience and NIHR Senior Investigator, The University of Nottingham), Elinor Griffiths (Research Grants Manager, University Hospitals Bristol NHS Foundation Trust); Simon Goodwin (Programme Manager, Central Commissioning Facility, NIHR); Alan Bradley and John Long (Patient and Public advisers).

**Contributors** SV, JB, SB, MB, RC, SC, KK, SP and CS involved in the conception and design of the study. SV, JB, JRB, SB, MB, RC, SC, TF, KK, LP, SP and CS made substantial contributions to development of the methodology and analysis. SV drafted the manuscript. JRB, TF, KK and LP led on the data collection and HT conducted the data analysis. All authors contributed to the critical revision of the manuscript for publication and approved the final version to be published.

**Funding** This report is independent research funded by the National Institute for Health Research (Research for Patient Benefit Programme, Home or hospital for people with dementia and one or more other multimorbidities: what is the potential to reduce avoidable emergency admissions?, PB-PG-0215- 36098).

**Data sharing statement** Supplementary files and the dataset are available from sarah.voss@uwe.ac.uk.

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
