## [Reviewer comments · BMJ Open]

ARTICLE DETAILS

TITLE (PROVISIONAL)	How do people with dementia use the ambulance service? A retrospective study in England: The HOMEWARD Project
AUTHORS	Voss, Sarah; Brandling, Janet; Taylor, Hazel; Black, Sarah; Buswell, Marina; Cheston, Richard; Cullum, Sarah; Foster, Theresa; Kirby, Kim; Prothero, Larissa; Purdy, Sarah; Solway, Chris; Bengner, Jonathan

VERSION 1 – REVIEW

REVIEWER	Laurie Coots Daras RTI International, USA
REVIEW RETURNED	02-Apr-2018

GENERAL COMMENTS	This paper addresses an important topic and is well written. I have only a few suggestions for the authors to consider:  -minor: the phrase "retrospective and observational" should have the word "observational" removed - if something is retrospective, it is implied to be observational -I didn't see a reference to the appendix -I understand there were no statistical comparisons, but in some places simple t tests would have been helpful. Also, given this was a small sample, I'm not sure how they got standard deviations on characteristics--perhaps describing how those were calculated will help the reader. -The tables and figures are misnumbered and have to be corrected -The first reference to Figure 2 is incorrect - and I think what follows the sentence "similar findings were evident for the provisional diagnosis as recorded by the attending clinician" are actually results not shown. If so, please update. Also, consider deleting all of the numbers if there's no accompanying table and just saying results are similar, as you had it. -Suggest breaking up the social circumstances and frailty section into two paragraphs so easier for the reader to see which refers to the figure and which refer to the table. -My main question is how was dementia defined? Did it distinguish severity? I would request the authors address that in the methods more clearly. Presumably it's documented but just as "dementia" or using ICD?
--

REVIEWER	Beverly Temple College of Nursing, Faculty of Health Sciences, University of Manitoba, Canada
REVIEW RETURNED	03-Apr-2018

GENERAL COMMENTS	Within the abstract and the paper - you use 999. I would suggest
--

	that if you are suggesting that this represents an emergency call - this should just be described as the numbers themselves are different across the world. The ethical access was not discussed - how the data was handled ethically was indicated. The methods described do not clearly what statistical tests were used to calculate the confidence intervals. you describe the descriptive results but indicate that some outcomes are 'more likely' etc. but not if you were considering your results statistically significant. (the results quoted all fall within the CI). there are two table 3s Quite a few of the references used are dated prior to 2010 and there has been a great deal of work done around dementia since that time - including some around emergency services. Chart audits are difficult studies and the limitations of these were not adequately discussed. The amount of missing data indicated is confusing within the tables and there is not adequate discussion of how missing data was addressed. in the discussion - you talk about how with increasing awareness of negative outcomes for people with dementia who are admitted - how does this lead to ambulance staff avoiding conveyance. this would need to be explained and I am not sure that this study could come to that conclusion.
--	--

VERSION 1 – AUTHOR RESPONSE

Reviewers' Comments to Author:

Reviewer: 1

Thank you for reviewing this submission and for your helpful comments. We have addressed each comment in turn as described below and revised the manuscript accordingly.

This paper addresses an important topic and is well written. I have only a few suggestions for the authors to consider:

Minor: the phrase "retrospective and observation" should have the word "observational" removed - if something is retrospective, it is implied to be observational - I didn't see a reference to the appendix - I understand there were no statistical comparisons, but in some places simple t tests would have been helpful. Also, given this was a small sample, I'm not sure how they got standard deviations on characteristics--perhaps describing how those were calculated will help reader.

The term "observational" has been removed from the manuscript. As stated at the end of the methods section, the analysis was exploratory, seeking to describe the nature of ambulance calls to patients with and without dementia, and therefore uses descriptive statistics only. Statistical hypothesis testing was not conducted due to the number of factors being considered and the risk of finding spurious statistically significant results. Moreover, as the clinical decisions made during call-outs are likely to be influenced by, for example, the crew, day of the week and time of the call, all of these unknown factors would need to be taken into account to make hypothesis testing valid. However, the sample size was large, so calculating standard deviations is appropriate for these data. We used recognised methods for calculating an estimate of the population standard deviation and this wording has been added to the manuscript.

The tables and figures are misnumbered and have to be corrected - The first reference to Figure 2 is incorrect - and I think what follows the sentence "similar findings were evident for the provisional diagnosis as recorded by the attending clinician" are actually results not shown. If so, please update. Also, consider deleting all of the numbers if there's no accompanying table and just saying results are similar, as you had it.

This has been corrected in the revised manuscript

Suggest breaking up the social circumstances and frailty section into two paragraphs so easier for reader to see which refers to the figure and which refer to the table.

This has been changed accordingly.

My main question is how was dementia defined? Did it distinguish severity? I would request the authors address that in the methods more clearly. Presumably it's documented but just as "dementia" or using ICD?

The data were coded according to whether dementia (and any other comorbidities) were noted in any section or free text area of the PCR. The coding template is lengthy and will be available as supplementary material on request. This has been clarified in the manuscript. Ambulance staff were not in a position to judge the severity of dementia.

Reviewer: 2

Thank you for reviewing this submission and for your helpful comments. We have addressed each comment in turn as described below and revised the manuscript accordingly.

Within the abstract and the paper - you use 999. I would suggest that if you are suggesting that this represents an emergency call - this should just be described as the numbers themselves are different across the world.

This has been changed in the revised manuscript.

The ethical access was not discussed - how the data was handled ethically was indicated.

The following paragraph has been added to the methods section of the manuscript: "HRA approval (IRAS: 202449) was obtained prior to data collection. Approval from an ethics committee was not required because the project involved the collection and analysis of retrospective and anonymised patient data only."

The methods described do not clearly what statistical tests were used to calculate the confidence intervals. you describe the descriptive results but indicate that some outcomes are 'more likely' etc. but not if you were considering your results statistically significant. (the results quoted all fall within the CI).

The following sentence has been added to the methods section of the manuscript: "95% confidence intervals for percentages/proportions were calculated using the Exact Binomial Method. 95% confidence intervals for means were calculated using standard methods assuming a normal distribution, which is appropriate given the large sample size."The manuscript states that hypothesis testing has not been carried out due to the large number of factors considered (see the response to reviewer 1).

There are two table 3s

This has been corrected in the revised manuscript.

Quite a few of the references used are dated prior to 2010 and there has been a great deal of work done around dementia since that time - including some around emergency services.

As part of the revision, four of the older references have been updated.

Chart audits are difficult studies and the limitations of these were not adequately discussed. The amount of missing data indicated is confusing within the tables and there is not adequate discussion of how missing data was addressed.

As the study uses retrospective data from ambulance Patient Care Records (PCRs), for some items the completion rates were low. However, the key items such as whether conveyance occurred and the length of call out had good completion rates. For most items the assumption was made that the

data were “missing at random”, and so those with missing data for the item in question were removed from the analysis of that item only. For this reason the number in the analysis (n) is different for each item reported. For the comorbidities, it was assumed that if the comorbidity had not been reported, then the patient did not have it, since there was no option to record “No” for the individual comorbidities in the PCR.

In the discussion - you talk about how with increasing awareness of negative outcomes for people with dementia who are admitted - how does this lead to ambulance staff avoiding conveyance. This would need to be explained and I am not sure that this study could come to that conclusion. This has been explained and clarified in the revised version.

FORMATTING AMENDMENTS (if any)

Required amendments will be listed here; please include these changes in your revised version:

- Kindly re-upload figures with at least 300 dpi resolution.

This is included with the revised submission.

VERSION 2 – REVIEW

REVIEWER	Laurie Coots Daras, PhD RTI International Waltham, MA, USA
REVIEW RETURNED	22-May-2018
GENERAL COMMENTS	All comments were adequately addressed - thank you.